# What factors enhance students' achievement? A machine learning and interpretable methods approach

Hui Mao[1,2]*, Ribesh Khanal[1], ChengZhang Qu[2], HuaFeng Kong[2], TingYao Jiang[3]*

1 School of Economics and Management, China Three Gorges University, Yichang, People's Republic of China, 2 School of Information Engineering, Wuhan Business University, Wuhan, People's Republic of China, 3 School of Computer and Information Technology, China Three Gorges University, Yichang, People's Republic of China

* sharot@163.com (HM); jiangty@ctgu.edu.cn (TYJ)

## Abstract

Prior research on student achievement has typically examined isolated factors or bivariate correlations, failing to capture the complex interplay between learning behaviors, pedagogical environments, and instructional design. This study addresses these limitations by employing an ensemble of five machine learning algorithms (SVM, DT, ANN, RF, and XGBoost) to model multivariate relationships between four behavioral and six instructional predictors, using final exam performance as our outcome variable. Through interpretable AI techniques, we identify several key patterns: (1) Machine learning with explainability methods effectively reveals nuanced factor-achievement relationships; (2) Behavioral metrics (hw_score, ans_score, discus_score, attend_score) show consistent positive associations; (3) High-achievers demonstrate both superior collaborative skills and preference for technology-enhanced environments; (4) Gamification frequency (s&v_num) significantly boosts outcomes; while (5) Assignment frequency (hw_num) exhibits counterproductive effects. The results advocate for: (a) teachers should balance direct instruction with active learning modalities to optimize achievement, and (b) early warning systems should leverage identifiable learning features to proactively support struggling students. Our framework enables educators to transform predictive analytics into actionable pedagogical improvements.

## 1. Introduction

Research on student achievement has constituted a fundamental area of inquiry in educational scholarship for decades, yet critical limitations persist in current methodological approaches. While numerous studies have investigated various aspects of learning outcomes [1–5], the predominant research paradigm continues to rely on examining isolated factors or simple bivariate relationships. This reductionist

**Data availability statement:** The data underlying this study are available in the dataset titled "Analysis of Student Learning Willingness", which is hosted on IEEE DataPort (https://dx.doi.org/10.21227/8h0p-fj20). The R language code used to run the OF model in this study has been made publicly available on Zenodo (https://doi.org/10.5281/zenodo.14788957). A screenshot of the operation and output of the OF model is also included as Supporting information.

**Funding:** This research was funded by the University-Industry Collaborative Education Program (No. 202102588008). The funders had no role in study design, data collection and analysis, decision to publish, or preparation of the manuscript.

**Competing interests:** The authors have declared that no competing interests exist.

approach fails to account for the complex, dynamic interactions between students' learning behaviors, instructional environments, and pedagogical methodologies that collectively shape academic achievement. Such oversimplification represents a significant theoretical and practical limitation, as it prevents a comprehensive understanding of the multifaceted nature of learning processes.

The emergence of educational artificial intelligence has introduced transformative potential for addressing these limitations. Recent applications of AI in education have demonstrated considerable success in solving fundamental teaching challenges, improving instructional management approaches, and enhancing learning outcomes [6–9]. Particularly noteworthy is the growing body of research utilizing student engagement data to model factors influencing academic achievement [10–12]. These developments reflect important advances, yet they remain constrained by their predominant focus on engagement metrics and their limited incorporation of broader instructional and environmental variables.

Three principal conceptual frameworks currently guide AI applications in education: data-driven learner analytics [13], computational learner modeling [14], and precision teaching methodologies [15–17]. These approaches collectively represent a paradigm shift toward evidence-based instruction, where analysis of behavioral data informs personalized learning interventions. However, despite these advances, significant gaps remain in both the scope of variables considered and the practical interpretability of findings for educators. Most existing studies either focus narrowly on learner characteristics or fail to provide actionable insights that teachers can readily implement in classroom settings.

This study addresses these critical gaps through an innovative methodological approach that combines the predictive power of machine learning with the explanatory clarity of interpretable artificial intelligence techniques. By applying five different machine learning models, namely Support Vector Machines (SVM), Decision Trees (DT), Artificial Neural Networks (ANN), Ordinal Forests (OF), and eXtreme Gradient Boosting (XGBoost), we analyze relationships among four key learning behaviors, six instructional environment variables, and final examination outcomes. Using final exam scores as our primary achievement metric, we simulate and interpret the complex relationships between these factors through explainable AI methods. This approach not only extends beyond conventional engagement metrics by incorporating a broader range of variables, but also ensures practical applicability for educators through transparent, interpretable results that can directly inform teaching practices and curriculum design.

The significance of this work lies in its dual contribution to both educational theory and practice. Theoretically, it advances our understanding of the complex interplay between multiple achievement factors. Practically, it provides educators with a data-driven framework for optimizing instructional design and implementing targeted interventions.

This study comprises six sections. Section 2 reviews relevant literature on achievement determinants and AI in education. Section 3 details our methodology using five machine learning models. Section 4 presents empirical findings on

factor relationships. Section 5 validates results through robustness checks. Section 6 discusses implications and offers evidence-based recommendations for educators.

## 2. Literature review

### 2.1. Selection indicators of student engagement

In Ubani's early prediction model [18], real data of students (high school, college, associate degree, bachelors and master's degree) were collected from schools in Nigeria, from January 2022 to March 2023, and there is a clear correlation between student scores and engagement levels. From the correlation heatmap, the correlation value of "writing score\reading score & total level" is 0.97, and the correlation between "math score & total level" is 0.92 (with a self-correlation value of 1).

Fidelia [19] modeled student engagement based on behavior through MOOCs platforms and used Pearson correlation to analyze the data, using datasets from 238 Canvas Network online courses. The study showed the association between students' achievement and student engagement through the student behaviors (i.e., "Distinct days active" and "Total number of posts made in discussion forums") of MOOCs platform ($r = 0.32$ to $0.38$).

Jasnas' experiment [20], conducted in the Biomedical University Study project with a blended-learning course, showed that using e-assessment to monitor student data continuously can predict the passing rate of the final practical exam. Gunawardena [21] used a multiple regression model to predict final exam scores for students with 3 learning procedural engagement data points from an in-person teaching course. In both studies, final exam scores were used as an indicator of academic outcomes, and it was tacitly assumed that student engagement data could predict these scores accurately.

A structural equation model was constructed by Natalia [22] which comprised student engagement data (such as time spent, completion of teacher-assigned tasks, time management, etc.) and achievement (scores in 4 courses) as variables, involving students from Compulsory Secondary Education (7th to 10th grade) from fourteen schools in northern Spain. The author used two absolute indices: the goodness-of-fit-index (GFI) and the adjusted goodness-of-fit-index (AGFI) in this paper. With a GFI of 0.980 and an AGFI of 0.964 ($p < 0.001$), the hypothesized model adequately represents the relationships in the empirical data. The results reveal that achievement is closely linked to engagement data.

Anna's research, especially during the first year of university, found that absenteeism can have detrimental effects on academic achievement, through an in-person teaching undergraduate Biology course [23]. Changing teaching methods, reducing absenteeism and improving student participation has significant positive effects on improving academic performance. Thomas' research [24] showed that attendance data typically show a strong positive relationship with students' achievement in an online learning environment. Jianzhong's study, which adopted a person-centered approach, investigated students' perceptions of teacher homework involvement using data from 823 9th-grade students in China. The study measured homework quality, autonomy support, and feedback quality, and the result showed that membership behavior was strongly associated with students' mathematics achievement [25].

Several scholars have conducted research on the relationship between students' academic performance and their engagement, and a consensus has emerged: (1) Student engagement is clearly linked with academic achievement. (2) Students' academic achievement is often directly measured by final exam scores, while student behavior data are often used to measure engagement, as demonstrated in papers above. Therefore, based on the previous studies, this paper intends to use final exam scores as a quantitative indicator of students' achievement, and use post-class assignment scores, class average scores, group activity and discussion scores, and attendance rates as engagement data to examine their impact on students' achievement.

### 2.2. Selection indicators of teaching environment and methods

Kaili's study investigated college students' overall levels of learning engagement and smart classroom preferences. A survey was conducted with a sample of students who had studied in this environment for one semester at a university in

central China. The results showed that higher level students had a stronger preference for smart classrooms [26]. Bing-zhuan's findings show that learning environment plays a significant role in stimulating students' interest and enhancing engagement [27], with 296 sophomores from six universities in China. Qun [28], taking an offline university art-design course as an example, proposed the idea that students' learning behaviors are frequently affected by teachers' in-class instructional approaches, and these approaches can effectively mirror the quality of teaching activities. Heethal's study evaluated the correlation between students' written test scores and tutor performance, concluding that students' written test scores were effective measures of tutor effectiveness [29], using the data of two groups of second-year medical students at MAHSA University Faculty of Medicine, Malaysia.

Many teachers have tried various teaching methods to promote students' enthusiasm to learn, such as using game-based methods [30,31], storytelling [32], videos [33] and other means as multimodal teaching approaches. But in the teaching process, teachers often face the challenge of balancing the time spent teaching core knowledge with the time allocated for classroom activities.

Based on the above literature, considering that teaching environment and methods may have an impact on students' achievement and their engagement, this paper adds indicators related to the teaching environment and methods, including "use of smart classrooms", "the frequency of using game-based teaching methods", "the frequency of using group activities or discussions", "the percentage of time spent on teacher explanations during class". These factors are considered as additional independent variables and their impact on students' achievement is discussed.

### 2.3. Existing limitations

While there is substantial research on the application of artificial intelligence in teaching, fewer studies focus on interpretable machine learning methods. Many studies rely on fitting student behavior data into models [18–22]. Although these articles explore AI technology modeling with engagement data, they often ignore the relationship between inputs and outputs. Moreover, it is evident that in practical applications, the more complex the model, the higher the "opaque box" nature of machine learning [34].

On the other hand, most literature on teaching methods focuses on the adoption of new multimodal teaching approaches, analyzing their effectiveness using subjective measures like questionnaires. However, this approach often lacks empirical evidence, and the persuasive power derived from real student data.

In view of these limitations, this study will be committed to making improvements in aspects such as indicator selection, model construction, and analysis methods, so as to explore the factors influencing students' achievement more deeply and provide more targeted and effective suggestions for teaching practice.

### 2.4. The impact of cross-regional factors on students' achievement/engagement

Based on the above research on the influence of various indicators on achievement, the following will further analyze the role of cross-regional factors on students' achievement/engagement, in order to comprehensively examine the system of influencing factors of students' achievement.

Shui-Fong's research aims to gain a thorough understanding of the contextual elements influencing student involvement, including cross-cultural viewpoints. The study examined the relationship between student engagement and factors such as grade, gender, and situational factors in 12 different countries/ regions. It also investigated whether engagement varied across countries with different levels of individualism and socioeconomic development. The survey included 3,420 students from 12 countries, who completed a questionnaire on their school involvement, teaching practices, and support from teachers, parents, and peers. The results from different regions show similar trends in student engagement at the same age, with no change across 12 different countries/ regions. Student engagement is positively correlated with situational factors such as teaching practice, teacher support and parent support [35]. In addition, the research literature on student engagement, achievement, final exam scores, student behavior, and teaching environments & methods

[18–29] includes studies from a range of countries, including Spain, Nigeria, Canada, the USA, Croatia, the UK, Macau/China, China, South Korea, and Hong Kong/China. Despite the diverse geographical contexts, these studies consistently highlight strong connections between students' achievement, engagement, and the teaching environments & methods employed. Therefore, we believe that studying effective ways to improve students' achievement/engagement has strong cross-regional applicability.

## 3. Methodology

Machine learning modeling method is the core method adopted in the paper. It involves learning from data and building a model by using different algorithms and technologies. This process includes steps such as data collection and processing, feature selection, and model selection, training, validation, and optimization. The ultimate goal is to enable the model to make accurate predictions or decisions on new, unseen data [18–22,34].

### 3.1. Research framework

The paper's research framework is illustrated in the following Fig 1.

First, we trained the model using different machine learning algorithms, incorporating various features (i.e., the 10 indicators described in Section 3.3) as input factors, and using final exam grades as the output label to measure students' achievement. Next, we compared the results and performance of the training models on test samples and identified the XGBoost model as the one with the highest accuracy. We then evaluated its performance and results. To ensure the model's correctness and robustness, we applied interpretability tools, such as Shapley Additive Explanations (SHAP), to enhance the transparency of the model's predictions and verify its validity.

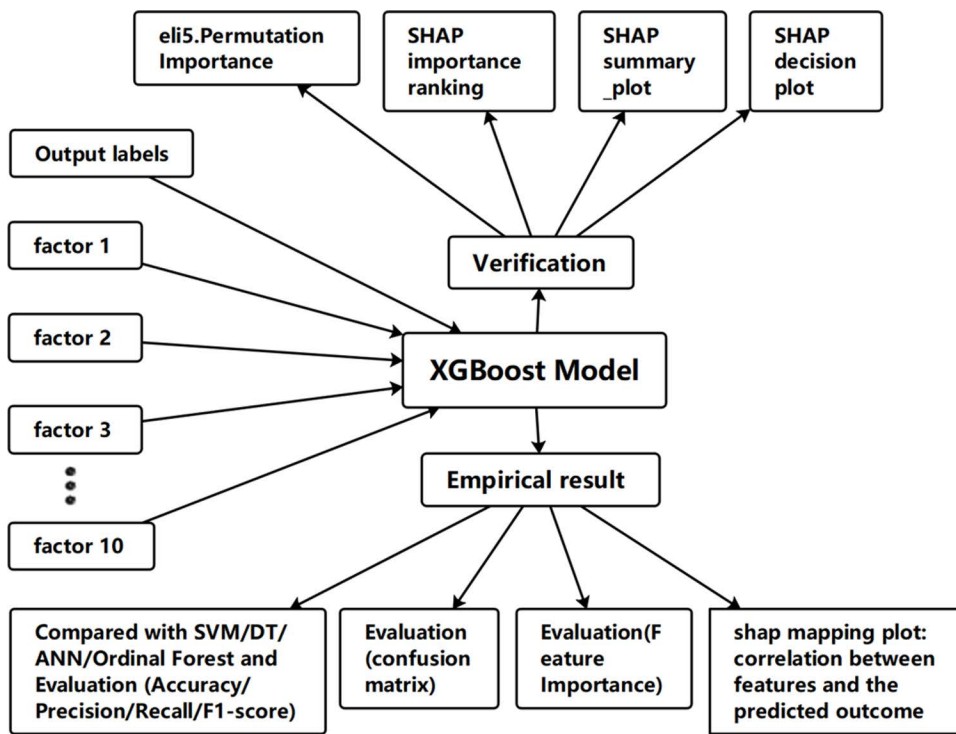

**Fig 1. The paper's research framework.**

## 3.2. Datasets

Theoretical courses, such as "Computer System," often face limitations in fostering meaningful learning interaction and student engagement due to a lack of practical components. In contrast, courses that incorporate hands-on experiments, such as building hardware or software, can more effectively enhance students' sense of achievement and engagement. Therefore, this paper chooses a pure theoretical course "Computer System" as an example to discuss the influencing factors and collect data from 10 indicators in the teaching process of the software engineering major of the School of Information Engineering for three consecutive years (2021, 2022, 2023). This course is an in-person teaching course offered to sophomore students majoring in computer science, and the assessment is in the form of a closed-book written exam at the end of the semester. This set of data serves as the foundation for constructing a student learning achievement classification model. Among the 29 professional courses in software engineering major, this course has the lowest pass rate in the final exam over the years. The data for this study was recorded by the micro-assisted teaching platform in the daily teaching process of the course "Computer System", involving 11 classes in 3 years, with a total of 449 students. The train_test_split () function was used to randomly divide the dataset into the training set and the test set proportionally. The total samples were 449, including 359 samples (80%) assigned to the training set and 90 samples (20%) to the test set.

## 3.3. Feature selection

The 10 basic indicators selected in this paper are shown in Table 1: (1) 4 student behavior indicators in class: attendance score (attend_score), average score of post-class assignments (hw_score), average score of group activities or discussions (discus_score), and average score of classroom question answering (ans_score). These students' behavior indicators are not only closely related to student engagement/achievement [18–25], but also can effectively reflect the quality and performance of teaching activities [28,29]. (2) 6 indicators of teaching environment and methods: use of smart classrooms (smart_cr), reflecting hardware quality of the teaching environment; frequency of game-based teaching methods such as storytelling and video viewing in a semester (s&v_num), which reflects the diversity of teaching methods; the frequency of using group activities or discussions in a semester (discus_num), the frequency of post-class assignments in a semester (hw_num), the frequency of classroom question answering in a semester (ans_num) and the percentage of class time spent on teacher explanations in a semester (explan_t), which reflect the teaching methods designed by teachers.

These 10 indicators are used as input variables, while students' final exam grades are used as the output label to measure students' achievement [18–22]. To monitor whether students pass or fail in a course, Ubani [18] grouped the students' performance into classes including 'Excellent' (A), 'Good' (B1, B2), 'PASS' (C, D), 'Fail' (E, F), providing references

**Table 1. Summary Table of Basic Indicators.**

| Type | Indicators | Definition | Reference |
|---|---|---|---|
| output labels (from final exam grades) | fe_tag | students' actual final exam scores are classified into three categories of failing/pass/good results (0, 1, 2); | [18–20,22,29] |
| Student behavioral data indicators | attend_score<br>hw_score<br>discus_score<br>ans_score | attendance score;<br>average score of post-class assignments;<br>average score of group activities or discussions;<br>average score of classroom question answering; | [20,23,24]<br>[20,21,25]<br>[20]<br>[20,21] |
| teaching environment & methods indicators | smart_cr<br>s&v_num<br>discus_num<br>hw_num<br>ans_num<br>explan_t | whether smart classrooms are being used for teaching;<br>the frequency of using gaming teaching methods such as storytelling and video viewing in a semester;<br>frequency of using game-based teaching methods such as storytelling and video viewing in a semester;<br>the frequency of post-class assignments in a semester;<br>the frequency of classroom question answering in a semester;<br>the percentage of class time spent on teacher explanations in a semester. | [26]<br>[30,32,33]<br>[28,29]<br>[21,25]<br>[21]<br>[28,29] |

for this paper. The final exam content consists of a total of 31 questions, covering the chapters learned in the semester. Scores are given according to students' answers, and the grading scale ranges from 0 to 100 points. The scores of the final exam are ultimately divided into three grades according to the school teachers' practice, based on certain score ranges: those below 60 points fall into Grade 0; those between 60 and 77 points (inclusive) are in Grade 1; and those with 78 points and above are in Grade 2, corresponding to failing, pass, and good grades, respectively. The data, from the micro-assisted teaching platform system, covers three complete semesters of teaching in 2021, 2022 and 2023.

### 3.4. Methods

In the same semester, teachers generally use similar teaching methods for the same course, with variations in teaching approaches typically occurring across different semesters. Due to this inherent consistency in pedagogical practices within a single semester, it is challenging to distinguish significant differences between individual student samples. Therefore, this paper considers various individual weak models in combination with integrated models, such as XGBoost, and compares their performance to identify the best model for building predictive models.

 **3.4.1. XGBoost model.** The XGBoost algorithm is a powerful Gradient Boosting algorithm created by Tianqi Chen et al. This ensemble algorithm excels in amalgamating underperforming classifiers to produce more precise models by synthesizing the results of all weak classifiers. As a result, XGBoost offers improved accuracy and greater resistance to overfitting compared to individual models [36,37]. For a given dataset, the first decision tree is used to fit the dataset, then the residuals (differences between actual and predicted values) are computed, and based on the residuals, the next decision tree is introduced and fitted. This process is iterated multiple times until the residuals reach an acceptable value. By adding up the fitting values of each decision tree, the final fitting result of the XGBoost algorithm can be obtained.

 While there is a limited amount of literature on the application of the XGBoost model in education, most research focuses on model fitting and multi-model comparison of student behavior and achievement data.

 For example, Amal adopted a variety of machine learning models to analyze student behavior during the learning process and proposed a prediction method based on the performance of different models (random forest, AdaBoost and XGBoost), evaluated their effectiveness. The experimental results show that compared with the original algorithm, the performance of the extensible XGBoost is better [38].

 Guangpeng used XGBoost algorithm to model students' learning behavior and estimate their achievements. In his paper, by comparing the traditional prediction method with XGBoost, he found an average improvement of approximately 8.4%, demonstrating the practical effectiveness of XGBoost in improving prediction accuracy [39].

 The above literature research shows that the XGBoost model has better fitting performance in the construction of a student learning model. The higher accuracy of the simulation means more accurate prediction results, which also means that the model is more suitable for application in education.

 The essence of the XGBoost algorithm is an improved version of the Gradient Boosting algorithm, and its adopted objective function can be represented as:

$$O_{bj} = \sum_{i=1}^{n} I(y_i', y_i) + \sum_{j=1}^{k} \Omega(f_j)$$

(1)

In Equation (1):
$O_{bj}$ represents the objective function; I is the loss function;
$\Omega$ is the regularization penalty function;
$y_i$ and
$y_i'$ are the true and predicted output values of the i-th sample in the dataset;
$f_j$ represents the output prediction value of the j-th decision tree; and
$n$ represents the count of samples.

In equation (1), the first term on the right side represents the loss function, which aims to train complex models that better fit the sample data. The second term represents the regularization penalty, which encourages simpler models to reduce the impact of noise disturbances. The objective function seeks to balance these two items between the loss function and the regularization penalty term to achieve the optimal effect.

For Boosting algorithms, the model from each iteration is preserved, and a new function is added at each iteration to improve the model's performance and decrease the objective function. By plugging in the fitted values after t rounds of iterations into the objective function, we can obtain:

$$O_{bj\,i}^{(t)} = \sum_{i=1}^{n} l[y_i, y'^{(t-1)}_i + f_i(x_i)] + \Omega(f_i)$$

(2)

In equation (2), $x_i$ represents the sample value. By using the Taylor expansion to approximate the loss function in the objective function above, we can obtain:

$$O_{bj\,i}^{(t)} = \sum_{i=1}^{n} [l(y_i, y'^{(t-1)}_i) + g_i f_t(x_i) + \frac{1}{2} h_i f_t^2(x_i)] + \Omega(f_i)$$

(3)

In equation (3), $g_i$ and $h_i$ are respectively the first and second derivatives of the loss function:

$$g_i = \partial_{y'^{(t-1)}_i} l(y_i, y'^{(t-1)}_i); \ h_i = \partial^2_{y'^{(t-1)}_i} l(y_i, y'^{(t-1)}_i).$$

Clearly, the training error of the above objective function depends only on the first and second derivatives of the loss function. With the model's iteration, the model optimizes between the loss function term and the regularization penalty term, ensuring a balance between model accuracy and overfitting resistance.

**3.4.2. Handling of ordered response variables.** In the existing literature research, there are already cases in various fields where different machine learning models rather than specialized methods and models for handling ordinal response variables are used to simulate and predict the classification of ordinal response variables. Haowei [40] adopted the XGBoost algorithm to train a prediction framework and predicted the influenza incidence rate as an ordinal variable classification. Abdelilah [41] used the extreme learning machine to classify the risk of fatal accidents on roads into four categories: high, medium, low, and safe. Wen [42] used six machine learning models to evaluate the deterioration and severity (severe/moderate/mild) of patients' diseases.

So far, there have also been many studies on the use of specialized methods and models for handling ordinal response variables [43–47]. The core idea is: Further, the responses are not nominal in nature. Instead, they are ordinal, since there exists a clear ranking relationship among the categories. For example, "Likely" doesn't imply it is twice of "Neutral" and "Very Likely" is three times of "Neutral". For such scenario the Logit model should be extended and is called Ordinal Logistic Regression. Ordinal logistic regression assumes that there is an underlying continuous variable among the categories, and each category corresponds to an interval of this continuous variable. By modeling this underlying variable, the probability of a sample belonging to each category is calculated. The sigmoid or logistic function can be used to compute the probability of falling into each category, and there are certain order constraints among the probabilities of the categories [45,46]. In the scenario of non - linear simulation, the ordinal forest model can be applied [48].

We noted that our output labels (0, 1, 2), which correspond to failing, passing, and good grades, represent an ordinal response variable. Therefore, we believe it's necessary to further investigate models specifically designed for ordinal data, like the Ordinal Forest. So we used a specialized model, the Ordinal Forest model in the R language, as described in [48],

for simulation and prediction, and compared the simulation accuracy with other models. Meanwhile, we added the function for handling ordinal response variables [45,46] to the XGBoost model with the best running accuracy to observe the simulation accuracy.

**3.4.3. Interpretable methods.** SHAP (Shapley Additive Explanations) is a Python package designed for interpretable machine learning, which is used to decipher the outcomes of machine learning models. Drawing inspiration from cooperative game theory, SHAP generates an additive explanation model, where all characteristics are regarded as "contributors" to the prediction. For each predicted sample, SHAP computes a value that represents the contribution of each feature to that specific prediction [49]. Let's assume that the i-th sample is represented as xi, the j-th characteristic of the i-th sample is xij, the model's forecasting for that sample is f(x$_i$), and the reference point for the whole model, which is typically the average of the target variable across all samples, is $y_{base}$. Shap_value satisfies the equation (4):

$$y_i = y_{base} + f(x_{i1}) + f(x_{i2}) + \ldots + f(x_{ik})$$

(4)

In contrast,
$f(x_{ij})$ symbolizes the SHAP value associated with
$x_{ij}$.Conceptually,
$f(x_{i1})$ represents the contribution of the first feature in the i-th sample to the ultimate prediction value
$y_i$. It signifies that the feature enhances the prediction value, resulting in a positive impact. Conversely, if
$f(x_{i1}) < 0$, it suggests that the feature diminishes the prediction value, leading to a negative impact.

Traditional feature importance only indicates which features are important but it does not clarify how the features impact the prediction results [50]. The greatest advantage of shap_value is that it reflects the influence of each feature in each sample, and it can also demonstrate the positive or negative impact. Based on shap_value, this method calculates and expresses the influence of each feature in each sample from the perspective of marginal contribution. Such influence can be displayed as positive or negative, a positive shap_value indicating a positive impact on the model's prediction, and a negative Shap_value indicating a negative impact. A shap_value of 0 indicates that the feature does not contribute to the model's prediction results.

The working principle of SHAP is to explain the impact of a certain feature taking a specific value by comparing the shap_value with the prediction when the feature takes a baseline value. Then, the sum of the shap_values of all features is used to explain why the prediction result differs from the baseline.

Tan [34] used the SHAP explainable framework to provide both global and local interpretations for 20 indices, such as daily frequency data from China's money, stock, and bond markets, as well as the foreign exchange and stability indices. By identifying the 8 most influential factors, this framework provides policymakers, regulators, and investors with important information about using valuable tools to improve risk resilience. Considering the success of SHAP in the economic field, this paper intends to apply SHAP to explain the influence of factors affecting learning engagement/achievement.

**3.4.4. Metrics for comparing model performance [34,51]. Accuracy** is a fundamental metric used to evaluate the performance of a classification model. It represents the proportion of correctly classified samples among all the samples. In other words, it gives an overall sense of how well the model is able to predict the correct classes for the given data.

**Precision** focuses on the positive predictions made by the model. It is defined as the proportion of true positive predictions among all the positive predictions. For example, in a spam - filtering system, precision tells you the proportion of emails that the system correctly identified as spam among all the emails it flagged as spam.

**Recall** measures the model's ability to correctly identify all the positive samples. It is the proportion of true positive predictions among all the actual positive samples. In a medical diagnosis context, recall represents the proportion of actually ill patients who are correctly diagnosed as ill by the model.

**F1_score** is a harmonic mean of precision and recall. It provides a balanced measure of a model's performance by considering both precision and recall.

## 4. Empirical results

This section consists of four parts: Comparison with other machine learning models; Building the XGBoost model (incorporating the handling of ordered response variable); Evaluating the performance and results of the XGBoost model; Using the shap_value map of the SHAP interpretable method to present the correlation between the predicted outcomes and the features in the "opaque box" of XGBoost model.

### 4.1. Comparison with different models

In this study, SVM, DT, ANN and Ordinal Forest (OF) models were used for comparison with XGBoost model (ignore/ add the handling of ordered response variable). We used the Ordinal Forest model in R language [48] for simulation and prediction, and implemented other models in Python 3.8.10. For each model, we evaluated the predictive efficacy of features on student learning achievement to ascertain whether the XGBoost model outperformed others.

Exhibited in Table 2 are the assessments of the test set's predictive performance across various models.

After the comparison, we can see XGBoost (incorporating the handling of ordered response variable) outperforms the other comparison models in various metrics, demonstrating that the order of models' predictive performance ranking is XGBoost (incorporating the handling of ordered response variable)> OF (Ordinal Forest) and XGBoost (ignore the handling of ordered response variable)> all the others.

From Table 2, we found that the prediction accuracy of Ordinal Forest model in R language [48] is better than that of the SVM, DT, and ANN models which do not consider the handling of ordinal response variable. Since its accuracy is equal to that of the XGBoost model which does not consider the handling of ordinal response variable, we thus added a function for handling ordinal response variables [45,46] to the XGBoost model. This time, we found the accuracy of the XGBoost model (incorporating the handling of ordered response variable) was significantly improved. Since the XGBoost model (incorporating the handling of ordered response variable) has demonstrated better prediction performance than other models, we still recommend the XGBoost model.

Because there is no significant difference in the data on teaching environment & methods between individuals in the same semester, the XGBoost model (add the handling of ordered response variable), which integrates multiple single weak models and ensemble algorithms, performed better than other machine learning models. This may be because the ensemble classifiers predictions are generally formed in parallel or iteratively by multiple weak classifiers, which are linear combinations of individual classifiers. Incorrect predictions are continuously corrected to achieve improved results.

**Table 2. Comparison of the prediction results.**

| Model Name | Accuracy | Precision | Recall | F1-score |
|---|---|---|---|---|
| SVM | 0.725926 | 0.798498 | 0.695617 | 0.718926 |
| DT | 0.777778 | 0.771750 | 0.764443 | 0.767545 |
| ANN | 0.829630 | 0.851846 | 0.821073 | 0.832432 |
| OF(Ordinal Forest) | 0.888889 | 0.896552 | 0.838710 | 0.866667 |
| XGBoost(ignore ordered response variable) | 0.888889 | 0.884175 | 0.879529 | 0.881567 |
| **XGBoost(incorporating ordered response variable)** | 0.922222 | 0.936111 | 0.907571 | 0.919111 |

## 4.2. Building the XGBoost model

Firstly, 10 indicators are used to measure teachers' multimodal teaching methods and student participation behavioral data, and students' achievement level (0, 1, 2) is used as the output label for establishing a prediction model. These levels correspond to three categories of final exam results: failing (0), pass (1), and good (2). Since the student data provided by the micro-assisted teaching platform is generally comprehensive, the workload of the original data pre-processing is minimal. Mainly the values of each column are normalized and numerically processed, and the drop_duplicates() function is used to remove duplicate rows.

Next, we employed the ordered_logistic_loss function in the XGBoost model to calculate the loss of the prediction for ordered response variables [45,46]. After optimization, the prediction accuracy of the model was significantly improved.

Regarding the division ratio of the training set and the test set in the model, both 8:2 and 7:3 are common choices in current literature. The ratio can be adjusted according to the actual data to balance the practical needs of model training and evaluation.[34,52–55] In the XGBoost model with the same parameter conditions, the train_test_split () function is used to divide the training set and test set respectively according to the ratio of 7:3 and 8:2. The performance of XGBoost model prediction results based on different ratios is shown in Table 3:

After comparing the performance of training and test sets split in a 7:3 ratio, it was found that an 8:2 split (training set: test set) yields more reliable and robust model evaluation results. Therefore, this paper uses the train_test_split () function to randomly divide the training set (359 samples, 80%) and the test set (90 samples, 20%) from a total of 449 samples.

Secondly, the model accuracy was improved by determining the optimal parameters through grid search and cross-validation. The goal of the Grid Search method is to find the optimal model parameters. It is a brute force parameter search method. By specifying different parameter lists, the exhaustive search is carried out, and the influence of each parameter combination on the model performance is calculated, so as to obtain the optimal parameter combination. This approach has the disadvantage of being time consuming, but it is still a reliable option in the face of complex models and multi-parameter situations. The average performance of the model on the test set is calculated through multiple iterations, so as to evaluate the generalization ability of the model [56]. This method helps reduce the risk of overfitting and under-fitting and makes the parameter selection more robust. The combination of grid search and cross-validation methods, such as GridSearchCV, can find the combination of parameters with the highest accuracy on the validation set within the specified parameter range. Although this method requires traversing all possible parameter combinations, which may be time-consuming in the case of large data sets and multiple parameters, it can ensure that the parameter with the highest accuracy is found within the specified parameter range. This approach has been proven effective and reliable in numerous studies [56–58].

The parameters employed to improve model accuracy are outlined below: (1) Gamma represents the threshold for further splitting of leaf nodes, with its default value is 0. (2) The n_estimators parameter determines the number of decision trees used in the model. Increasing the value of n_estimators allows the model to learn more features and patterns, thereby improving the model's predictive performance. However, increasing the value of n_estimators may also increase the complexity of the model, potentially leading to overfitting. (3) A higher max_depth allows the model to capture more detailed and localized patterns within the samples. (4) Subsample indicates the sampling rate of the training sample, with its default value is 1. Reducing the subsample value makes the algorithm more cautious, helping to prevent overfitting. (5) Colsample_bytree, representing the column sampling rate, with a default value of 1. (6) The regularization parameter

**Table 3. Comparison of the prediction performance.**

| XGBoost Model | Accuracy | Precision | Recall | F1-score |
|---|---|---|---|---|
| Ratio of 7:3 | 0.874074 | 0.873631 | 0.851749 | 0.861333 |
| Ratio of 8:2 | 0.922222 | 0.936111 | 0.907571 | 0.919111 |

**Table 4. Optimal Parameters.**

| Parameters | Range of parameter | Optimal value |
|---|---|---|
| learning rate | [0.3, 0.1, 0.01] | 0.1 |
| max_depth | 3-10 | 4 |
| min_child_weight | 1-6 | 2 |
| n_estimators | 0-100 | 20 |
| gamma | [0,0.1,0.2,0.3,0.4,0.5] | 0.4 |
| subsample | [0.6,0.7,0.8,0.9,1] | 0.6 |
| colsample_bytree | [0.6,0.7,0.8,0.9,1 | 0.7 |
| reg_lambda | [1e-5,1e-4,1e-3, 1e-2, 0.1, 1, 100] | 1 |

Reg_lambda, which uses L2 regularization and defaults to 1, regulates the influence of model complexity. Increasing the reg_lambda value reduces the likelihood of model overfitting. (7) Learning rate determines the weight of new features after each iteration. It can prevent overfitting by making the computation process more conservative, with a default value of 0.3.

In Table 4, the parameters that require optimization in the XGBoost model, the range in which parameter optimization occurs, and the optimal parameters are all presented in detail.

Finally, we summarize the adjusted parameters and proceed with the model prediction.

### 4.3. Performance of XGBoost model

The assessment of the prediction model centers on appraising the performance of the XGBoost model (incorporating the handling of ordered response variable) and ascertaining its predictive proficiency.

A.      Accuracy/Precision/Recall/F1_score

In XGBoost model, the testing set's accuracy, precision, recall, and F1_score were evaluated at 0.922222, 0.936111, 0.907571, and 0.919111. Considering that data are constrained by the fact that student behavior data and teaching environment & methods data tend not to differ significantly between individuals in a semester, this accuracy shows that the XGBoost model has strong predictive performance.

B.      XGBoost model's confusion matrix

Fig 2 illustrates a confusion matrix generated by the XGBoost model, serving as a visual tool for evaluating the effectiveness of classification. In this matrix, each column corresponds to the model's predicted values, and every row signifies the actual samples' values. It is worth noting that the entries along the diagonal indicate the count of accurately classified samples [34].

According to Fig 2, out of 90 samples in the test set (data split into training and test sets in an 8:2 ratio), 14 good learning achievement samples, 42 pass samples, and 27 failing samples were accurately classified. This result in a sample classification accuracy of 92.22%, demonstrating that only a few samples were misclassified into neighboring categories. There were 2 misjudged samples in the good-level class, 1 in the pass-level class and 4 in the failing-level class, amounting to a total of only 7.88% misclassified samples. This supports the conclusion that the XGBoost model performs well in predicting student learning achievement.

This time, the error rate of good-truth labels (12.5%) is higher than that of pass-truth labels (2.33%), which is mainly due to the low proportion of high-level samples. When the number of samples in the dataset is small, the model may tend to predict the sample to be the larger class label. If the sample size of a particular class is relatively small in the total sample, the model may encounter challenges in dealing with this class, because there are fewer samples of this class in the training data, and the model may not perform as well in this class as on the high frequency class [59]. In this case, even

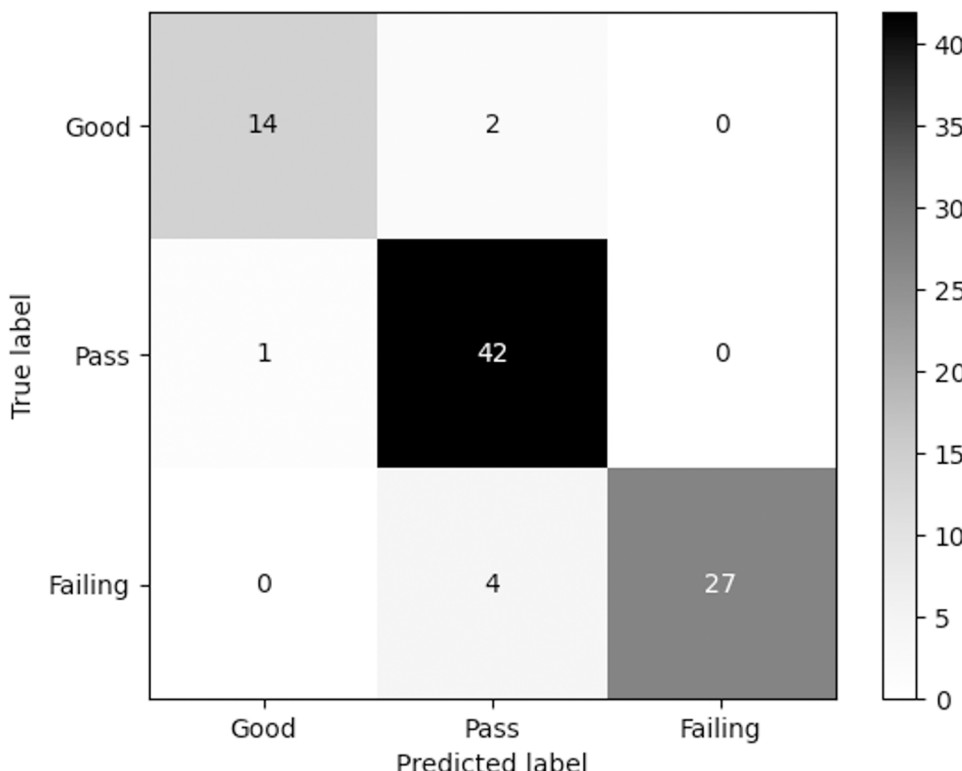

**Fig 2. Confusion matrix of XGBoost model.**

if the overall performance of the model is good, the ability to handle low-frequency categories may be weak, resulting in a relatively high error rate for good-truth labels (i.e., low-frequency categories).

C. XGBoost models Feature Importance ranking

The function of feature importance ranking in the XGBoost model clarifies the impact of various variables and highlights key functionality.

The XGBoost model comes with a function for plotting the importance of features. The function plot_importance can directly output a graph showing the ranking of feature importance in the XGBoost model, as shown in Fig 3 below:

We can see from Fig 3 that the features ranked higher have a greater impact on the judgment result than those ranked lower, indicating that the former are more important than the latter. The top 6 significant features are hw_score, ans_score, discus_score, explan_t, attend_score, and smart_cr. This result is consistent with the teachers' experience in theory teaching, which indicates the validity of XGBoost model in predicting students' achievement.

## 4.4. An "inside" analysis of the results: Correlations and safety point

In this section, the shap_value mapping plots are used to represent the correlation between the predicted outcome and various features in the "opaque box" of the XGBoost model. For the first six significant features of the XGBoost model (Fig 3), corresponding shap_value mapping plots (Fig 4A–4F) are presented to help understand the model prediction process. Shap_value is represented on the vertical axis of the shap_value map.

As depicted in Fig 4A–4F, the vertical axis on the left represent shap_value, and the horizontal axis represents the spectrum of the individual feature's value, respectively. The commonality of these six important features is that with

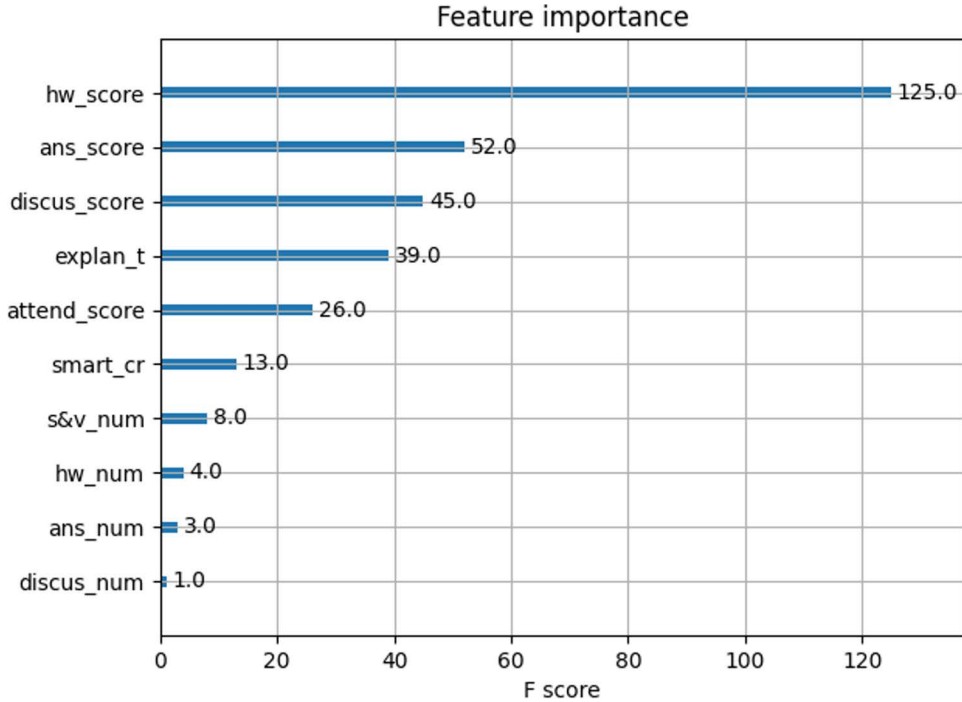

**Fig 3. XGBoost-importance feature importance ranking chart.**

the increase of the feature value, shap_values will increase, and the tendency of the sample to be predicted as good achievement status will also increase. A horizontal line passing through a point on the graph where the shap_value is 0 is considered a safety point. A vertical line through this point divides the entire graph into two intervals: the pass-failing achievement zone and the good achievement zone. In this paper, by introducing the feature safety points, teachers can identify the key groups that need attention in time.

Here we will provide a detailed explanation of the shap_value mapping plots for the above 6 features.

**4.4.1. Correlation between hw_score, ans_score, discus_score, attend_score and the predicted outcome.** In Fig 4A, we can see: (1) for a sample with a hw_score in the range [0,0.84], (normalized to a percentage system, where 0.84 corresponds to an 84% average assignment score), the shap_value is negative. This indicates that students in this range are more likely to be predicted as having failed to pass learning achievement. These students represent a group that requires greater attention and support from teachers. (2) When hw_score falls within the range of [0.84,1], it indicates that shap_value is positive, indicating that the sample is more likely to be predicted as having good learning achievement. In this example, a normalized assignment score of 0.84 is the safe point reference value for teachers in this model to predict in advance which students are likely to have pass/failing achievement outcomes (Note: Reference values may vary depending on the input of different data sets). At the same time, the positive correlation trend between the value of hw_score feature and shap_value (i.e., the prediction classification outcome) is also presented and verified through the graph sample distribution.

Similarly, as shown in Fig 4B–D, ans_score, discus_score, attend_score and the predicted outcome have similar connections as hw_score. The graph sample distribution presents and verifies the positive correlations between ans_score, discus_score and attend_score feature values and shap_values, namely, the prediction classification outcome. Only the safety point reference values corresponding to each feature are different. For example, the safety point reference values of ans_score feature,

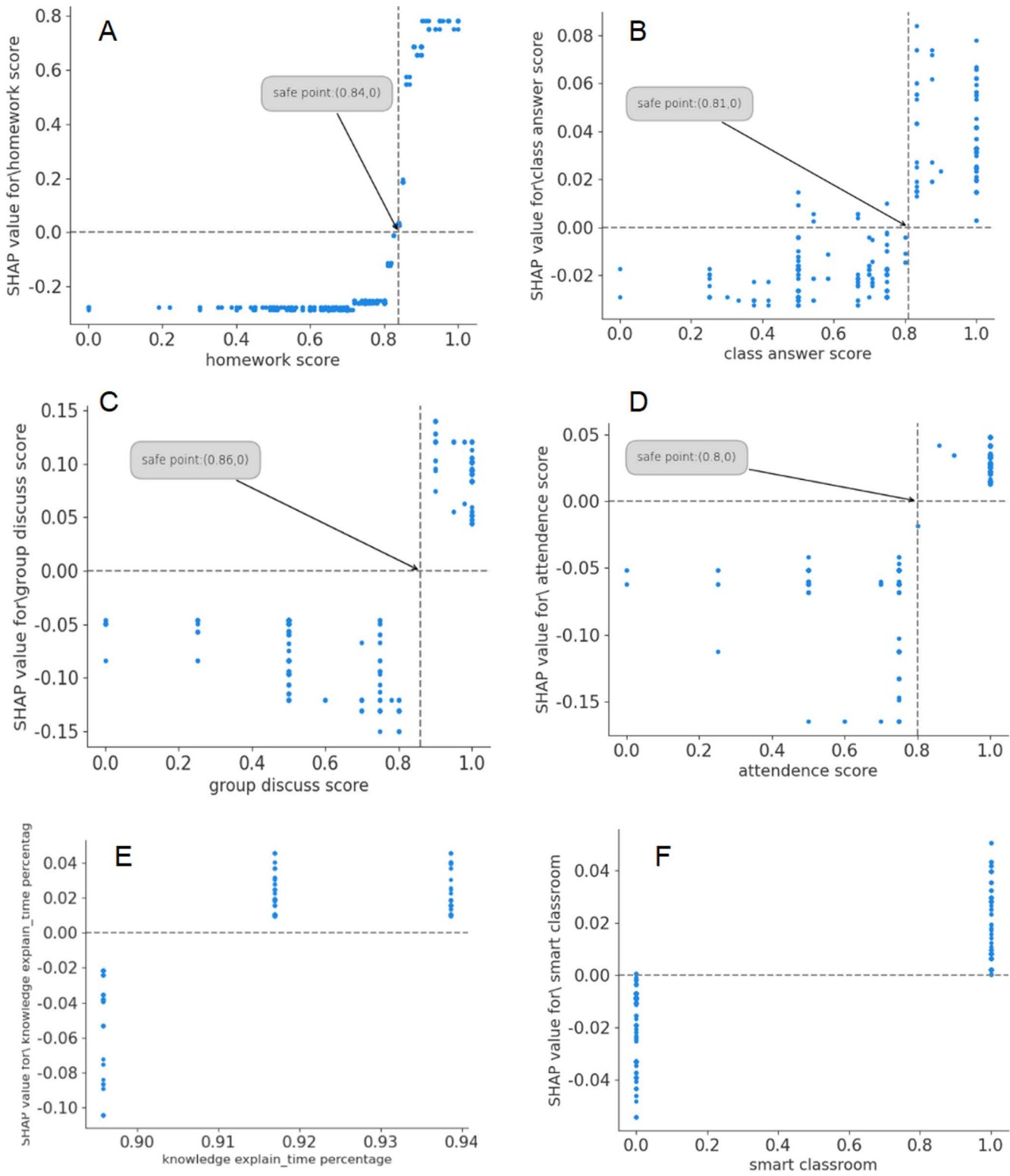

**Fig 4. The shap_value mapping plots for the top 6 features.**

discus_score feature and attend_score fessature are 0.81, 0.86 and 0.8 respectively. The samples of students who fall into the pass/failing achievement region below the reference value of each feature are the group that needs more attention from teachers.

Overall, there is a positive correlation between hw_score, ans_score, discus_score, attend_score and the predicted outcome. The finding suggests appropriate measures should be taken to encourage students to participate in homework, class questions, group activities or discussions with the highest quality possible, attract students to class and increase attendance, which will effectively improve students' final achievement.

**4.4.2. Correlation between explan_t & smart_cr and the predicted outcome.** The value ranges of the feature variables explan_t and smart_cr are relatively narrow. Students in the same grade usually have the same explan_t value and smart_cr value, because the teaching environment & methods used by teachers in the same semester are often the same. As a result, the shap_value mapping graph for these two features shows a clear block distribution, while their trends can be analyzed through the shap_value mapping graphs (Fig 4E, F). As can be seen from the shap_value mapping plot for explan_t, in this case, basic teacher time must be guaranteed to explain key knowledge in order to maintain student learning status (When the percentage of class time devoted to explaining key points >= 90%, the shap_value is positive, which means high probability of good achievement; When it is less than 90%, shap_value is negative, and the trend is opposite). The correlation between smart_cr and the predicted outcome shows that the use of smart classroom (assuming good teaching environment and conditions) can enhance students' learning achievement.

The safety point reference value obtained in this example is closely related to the grading habits of individual teachers, so it is only for reference, but it can provide guidance for teachers in actual teaching practice.

## 5. Verification

The verification is carried out at two levels: features and samples. Among them, the verification at the feature level includes the use of SHapley Additive exPlained (SHAP) interpretability tool to compare and verify the correctness of XGBoost model feature importance conclusions and using eli5.PermutationImportance method to forecast under disturbing single row data, to evaluate the robustness of the model. At the sample level verification, SHAP was used to respectively explain and verify the correctness of the model results through the overall samples and individual samples.

### 5.1. Verification at features level

A. Replacing the sorting method of feature importance

The test uses the Shap_summary method instead of XGBoost's feature importance ranking method to compare and validate feature importance ranking results from a contribution margin perspective.

By comparing the feature importance of the XGBoost model (Fig 3) with the shap_value_importance in Fig 5, we can find that the overall ranking of the influence of 10 characteristics on the prediction results is basically the same. This shows that the output result of XGBoost model is robust.

As we can see from Fig 5, the factors that have a higher marginal contribution to determining good/pass learning achievement are as follows: hw_score, discus_score, ans_score, attend_score, explan_t, smart_cr and s&v_num. Similarly, the factors with higher marginal contribution to failed learning achievement are the same set of features, with a slightly different order of importance. For different categories, the marginal contribution importance order of each factor is slightly different. Among them, students with good/pass learning achievement usually achieve relatively high scores in group discussions or activities and are more inclined to smart classrooms with better teaching hardware conditions.

B. *Disrupting single column data*

To verify the robustness of the XGBoost model against abnormal data and validate the importance of each feature of XGBoost model predicted results, this paper uses eli5. Permutation Importance method validation. Its working principle is

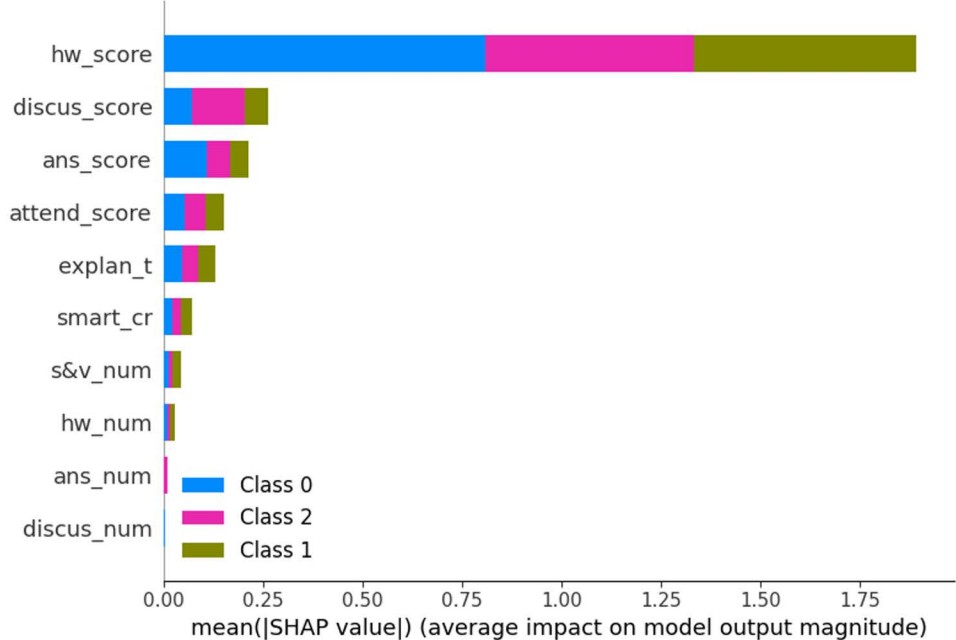

**Fig 5.** shap_value_importance figure.

**Table 5. Feature importance ranking of disrupting single column data.**

| Weight | Feature |
|---|---|
| 0.4637 + -0.0501 | hw_score |
| 0.0059 + -0.0059 | s&v_num |
| 0.0059 + -0.0059 | smart_cr |
| 0.0059 + -0.0059 | attend_score |
| 0.0044 + -0.0073 | explan_t |
| 0.0030 + _0.0073 | discus_score |
| 0.0030 + _0.0073 | ans_score |
| 0 + -0.0000 | ans_num |
| 0 + -0.0000 | discus_num |
| 0 + -0.0000 | hw_num |

to randomly shuffle the data values of any feature to destroy the corresponding relationship of the original data, and then examine the change of the model prediction effect. At this time, the difference in the performance of the model can reflect the robustness of the model, and also reflect the importance of the disrupted data columns. After that, the data of the scrambled column is recovered, and the same scrambled data operation is repeated on the next column until the importance of each column is calculated and the sorting is complete. The results of this verification are shown in Table 5.

It can be seen from Table 5 that through the single-column data disruption method, the importance weights of each factor's influence on the model prediction results are ranked as follows: hw_score, s&m_num, smart_cr, attend_score, explan_t, discus_score, and ans_score. The impact weights of other factors, ans_num, discus_num, and hw_num, are negligible. It can be seen that except for s&v_num feature, the influence ratios of other features is basically consistent with the analysis of the XGBoost-importance feature ranking diagram (Fig 3). This result means that the conclusions of the

XGBoost model show better robustness even when the data of a certain feature is randomly scrambled. At the same time, we also note that the s&v_num feature, which represents the teaching methods of games such as storytelling and video viewing, can't be ignored from the perspective of disrupting the single-column data approach.

## 5.2. Verification at samples level

This section verifies the model results and makes further interpretation of the model results at the sample level using the SHapley Additive exPlained (SHAP) interpretability tool [60].

### A. *using summary_plot for total samples*

We use SHAP summary_plot to describe the overall correlation between students' achievement of all samples and each feature, and use the calculated shap_value of each feature of each sample to help understand and verify the importance ranking results.

Considering the training set sample with 10 fundamental input features and an output label representing the students' achievement variable with 3 categories (failing, pass, good states), we use the SHAP summary_plot that can clearly describe the association state of all samples with each feature. In Fig 6, when the shap_value associated with the value of feature j for Xi is positive, it suggests that the contribution of feature j to the inclination of Xi towards good learning achievement is positive. The SHAP summary_plot is shown in Fig 6.

Fig 6 serves to comprehensively rank the impact of features on the prediction model, with the features listed on the left side of Fig 6, and their values indicated by colors ranging from blue to purple to red on the right. The horizontal axis depicts the shap_values, which quantify the scope and scale of shap_values for each feature across the samples. Every row signifies a feature, and each point symbolizes a sample, maintaining uniformity in the sample size across each row (449 sample points per row). Considering the initial row in Fig 6, hw_score holds the highest rank, identifying its substantial contribution

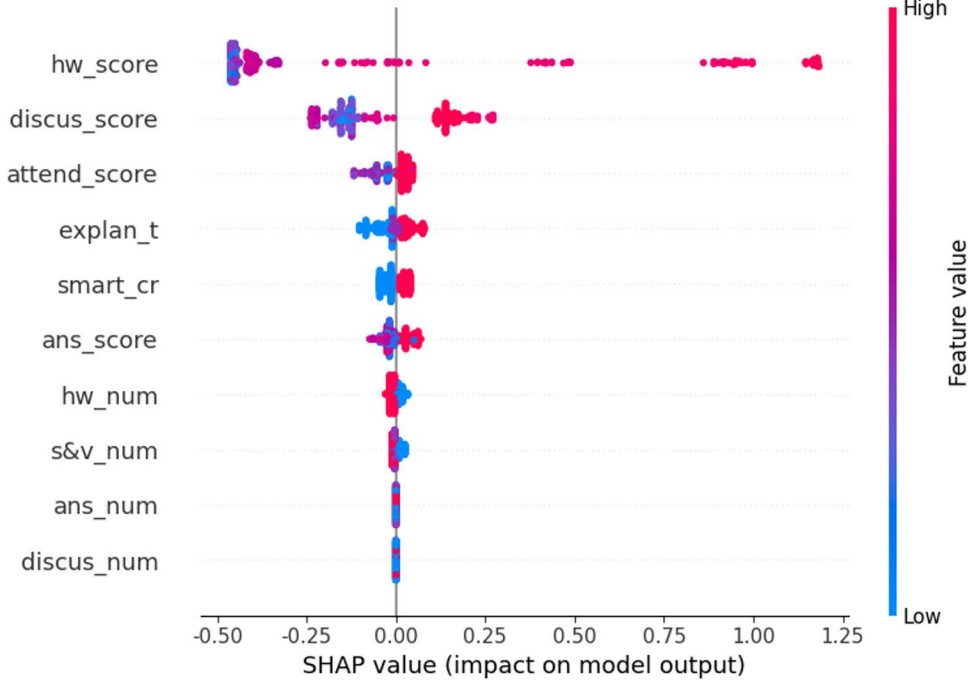

**Fig 6. SHAP summary_plot.**

to the prediction model. In addition, a smaller hw_score value (blue area) corresponds to a negative shap_value, meaning that a sample with a lower value is more likely to be predicted to have a higher probability of students failing in their achievement. With the increase of hw_score (purple and red areas), shap_values gradually become positive, indicating that samples with higher hw_score feature values are more likely to be predicted to have good achievement levels for students. And with the increase of hw_score feature values, achievement predictions are steadily improved.

In addition, discus_score, attend_score, explan_t, smart_cr, and ans_score also exhibit similar patterns, showing a left-blue, middle-purple, and right-red feature in the plots. A deeper red color indicates a larger indicator value, and a shap_value flagged positive can elucidate a good achievement inclination.

On the other hand, it can also be clearly seen from Fig 6 that hw-num feature interestingly presents the opposite result, showing the characteristics of red on the left and blue on the right, reflecting that an excessive number of assignments, which may have a negative effect on students' achievement.

However, the influence of s&v_num feature on results is vague and minor, and ans_num and discus_num show that they have little influence on the shap_value and predicted results, and shap_value is basically around 0. Therefore, from the analysis of the overall sample level, it can be said that factors such as s&v_num, ans_num, and discus_num have less impact on students' achievement.

This method further confirms the correctness of the feature importance ranking results of XGBoost model from the overall sample level.

C.    ***Using decision plot for individual sample***

In a single sample, it is feasible to discern the incremental contribution of each feature. We use SHAP decision plots to provide local interpretation of individual samples, as shown in Fig 7, where samples 39–50 from the training set are randomly selected for explanation.

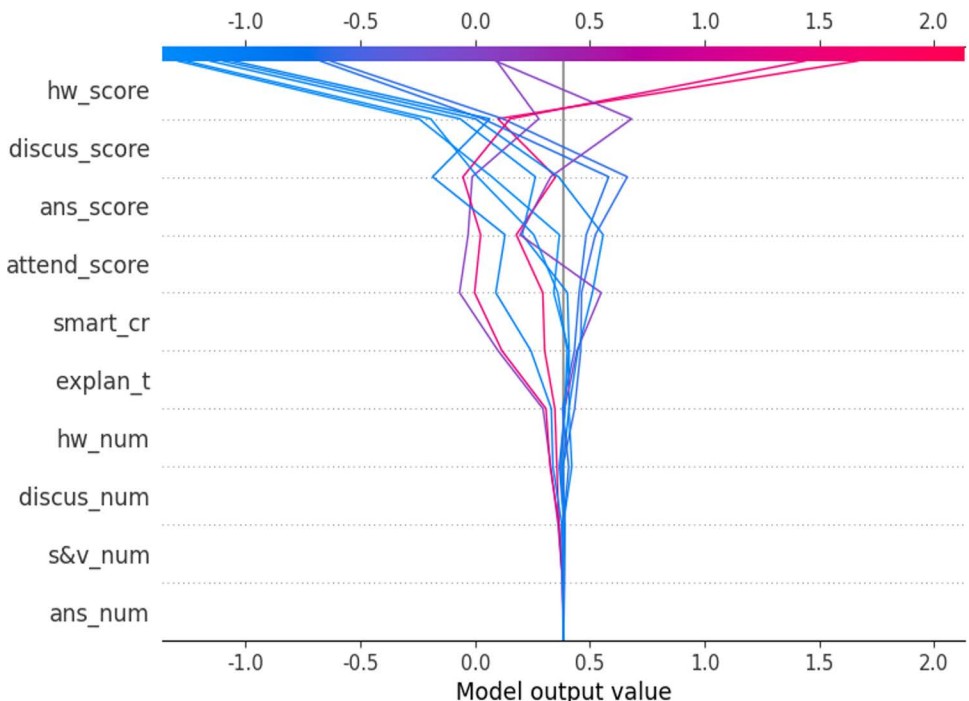

**Fig 7. SHAP decision plot for samples 39-50.**

The decision plot can display the impact of features on the results for multiple instances in one graph. In general, features located higher up have a greater impact. In this case, the X-axis describes the direction and strength of each feature's pull on the final predicted value.

In the SHAP decision plot, the gray vertical line marks the model's baseline value, while the colored lines represent the prediction, indicating whether each feature moves the output value higher or lower than the average prediction. The feature values are shown next to the prediction line for reference. Starting from the bottom of the graph, the prediction line shows how the shap_value accumulates from the baseline to the top of the plot, representing the predicted score of the sample. For the influence of each feature of each sample on the shap_value of the sample, the trend can be seen through the generated curve, and finally the predicted score of the sample can be accumulated. The red curve indicates that the cumulative effect of shap_value on a single sample is positive, while the blue curve indicates that the cumulative effect of shap_value on this sample is negative.

From the pull direction and strength of each curve of the SHAP decision graph in Fig 7, it can be seen that for sample individuals in samples 39–50: hw_score is the feature that contributes most to the prediction results, followed by discus_score, ans_score, attend_score, smart_cr, explan_t, and hw_num. These characteristics together determine the result of the final state of the sample. The other three features, discus_num, s&v_num and ans_num, have little impact on samples. Similarly, this conclusion is basically consistent with the feature importance ranking of XGBoost model. This method intuitively confirms the correctness of the conclusion of XGBoost model from the level of individual samples.

## 6. Discussion, conclusion, and implications

### 6.1. Discussion

This study introduces innovative methods to simulate and analyze the factors that enhance students' achievement. Through modeling and analysis, several important insights into teaching practices have emerged:

(1) There is a positive correlation between hw_score, ans_score, discus_score, attend_score and the predicted outcome. This means that appropriate measures of teaching design should be taken to encourage students to participate in homework, class questions, group activities or discussions with the highest quality possible, and to attract students to class and increase attendance, which will effectively improve students' final achievement. This further validates the research conclusion in literature [18–23,25]. (2) Students who achieve higher learning outcomes tend to score better in group discussions or activities and show a preference for smart classrooms equipped with superior teaching hardware. Through the analysis of actual data, we conducted further verifications and arrived at research conclusions that are extremely similar to those in Reference [26]. (3) We also found that the s&v_num feature, which represents the teaching methods of games such as storytelling and video viewing, can't be ignored. This indicates that the research on gamified teaching methods in References [30–33] has practical significance in improving students' learning outcomes. (4) Interestingly, hw-num feature presents the opposite impact, reflecting that an excessive number of assignments may have a negative effect on students' achievement. The importance of "the quality of assigned homework" is far greater than that of "the number or frequency of assigned homework", which further deepens the research content of Reference [25]. Similarly, factors such as the frequency of group activities or discussions and the frequency of answers in class have less impact on students' achievement. Teachers may pay less attention to their frequency and quantity. (5) It is necessary for the teacher to plan the time proportion of explaining knowledge points. If the degree of knowledge acquisition of students in the classroom is not guaranteed, it will also affect the achievement of students. This conclusion further validates and deepens the view of "effectiveness" in Reference [29]. (6) Introducing safety points for features can help teachers identify students or groups in need of additional support or attention. These findings support balancing instructional modalities and data-driven early warning systems. Building on these findings, future research will further validate and refine the framework through ongoing teaching practices, offering direct insights for curriculum design and classroom application.

## 6.2. Conclusion

Based on the empirical results and validation, the following conclusions can be drawn:

First, in comparison to other machine learning models like SVM, DT, ANN, and Ordinal Forest, the performance of the XGBoost model is notably superior to other classifiers, exhibiting greater accuracy and more robust generalization capabilities.

Second, using XGBoost in combination with SHAP for further analysis, we found that 4 indicators related to students' behavior (hw_score, ans_score, discus_score, and attend_score) have the greatest impact on students' achievement. In addition, ensuring adequate time to explain knowledge points, adopting good teaching environments and conditions, and using game teaching methods such as storytelling and video-based learning significantly contribute to student success. On the contrary, the frequency of homework, the frequency of group activities or discussions and answers have little influence on the prediction results. Overall, it is feasible to stimulate and interpret the data-driven early warning system.

This study has some limitations. First, the selection of indicators in this study may not be comprehensive enough. We only considered explicit data related to student participation and teaching methods, while ignoring pre-existing student interests, teacher delivery style (e.g., tone of voice, gestures), and classroom environmental variables. Further research could add classroom-related environmental variables to measure their impact more accurately. Second, the modeling and analysis in this paper are built upon data collected over three semesters. Future research should incorporate more data from subsequent semesters to further optimize the simulation and improve model accuracy.

## 6.3. Implications

The findings of this study offer valuable insights for improving the management of the teaching process:

Firstly, by introducing the interpretable method combined with machine learning modeling, this study has effectively simulated, explained and verified the influence of various factors on students' achievement. As demonstrated in the case studies presented in this paper, combining teaching data with machine learning models and interpretable methods not only ensures the accuracy of simulations but also provides a clear explanation of their underlying mechanisms. This approach is highly applicable to teaching process management, as it enables educators to gain a deeper understanding of how different factors affect student outcomes.Secondly, the research highlights the combination of artificial intelligence technology and teaching process management, emphasizing its practical significance. In addition, this paper expands the literature on the application of artificial intelligence technology in education. In future research, similar methodologies could be applied to other computer science courses that involve experimental components, building on the conclusions drawn in this study to enhance teaching practices across various disciplines.

## Acknowledgments

We are thankful to the anonymous reviewers and journal editors for their efforts in the evaluation of this study. We are also thankful to Associate Professor Thomas Stephen Ramsey for his support in editing the language of this manuscript.

## Author contributions

**Conceptualization:** Hui Mao, TingYao Jiang.

**Data curation:** Hui Mao.

**Funding acquisition:** HuaFeng Kong.

**Methodology:** Hui Mao, ChengZhang Qu.

**Software:** ChengZhang Qu.

**Supervision:** TingYao Jiang.

**Writing – original draft:** Hui Mao.

**Writing – review & editing:** Ribesh Khanal, HuaFeng Kong, TingYao Jiang.

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
