## [Decision Letter · Decision Letter 0]

3 Jul 2024

PONE-D-24-23837What Factors Enhance Student Learning Willingness? A Machine Learning and Interpretable Methods ApproachPLOS ONE

Dear Dr. Mao,

Thank you for submitting your manuscript to PLOS ONE. After careful consideration, we feel that it has merit but does not fully meet PLOS ONE’s publication criteria as it currently stands. Therefore, we invite you to submit a revised version of the manuscript that addresses the points raised during the review process.

We look forward to receiving your revised manuscript.

Kind regards,

Chun-Yen Chang

Academic Editor

PLOS ONE

Journal Requirements:

2. Please note that PLOS ONE has specific guidelines on code sharing for submissions in which author-generated code underpins the findings in the manuscript. In these cases, we expect all author-generated code to be made available without restrictions upon publication of the work. 

Please review our guidelines at https://journals.plos.org/plosone/s/materials-and-software-sharing#loc-sharing-code and ensure that your code is shared in a way that follows best practice and facilitates reproducibility and reuse.

"This research was funded by the University-Industry Collaborative Education Program (No. 202102588008)."

4. Please note that funding information should not appear in the Acknowledgments section or other areas of your manuscript. We will only publish funding information present in the Funding Statement section of the online submission form. Please remove any funding-related text from the manuscript.

**Additional Editor Comments:**

Dear Authors,

Several areas need improvement and clarification to enhance the quality and impact of the manuscript. I'll call your attention to the most substantive of their concerns:

1. Introduction Clarity:

- Clearly define primary AI models in teaching.

- Justify using final exam scores as a reflection of learning willingness.

- Include geographical context in literature review to highlight cultural differences.

2. Methodological Issues:

- Provide a clear research framework, visualizing model roles and inputs/outputs.

- Explain and cite reasons for categorizing exam scores and using specific predictors.

- Address limitations like equating exam results with learning motivation, and consider adding self-efficacy measures.

3. Model Parameters and Evaluation:

- Clarify the need for changing model parameters after cross-validation.

- Justify dataset split ratios and provide sample details.

- Structure results to evaluate and compare model performance (e.g., XGBoost vs. SVM, DT, RF, ANN) with detailed validation methods and experimental values.

Reviewers' comments:

Reviewer's Responses to Questions

**Comments to the Author**

1. Is the manuscript technically sound, and do the data support the conclusions?

Reviewer #1: Partly

Reviewer #2: Yes

Reviewer #3: Partly

2. Has the statistical analysis been performed appropriately and rigorously? 

Reviewer #1: Yes

Reviewer #2: Yes

Reviewer #3: Yes

3. Have the authors made all data underlying the findings in their manuscript fully available?

Reviewer #1: Yes

Reviewer #2: Yes

Reviewer #3: Yes

4. Is the manuscript presented in an intelligible fashion and written in standard English?

Reviewer #1: No

Reviewer #2: Yes

Reviewer #3: Yes

5. Review Comments to the Author

Reviewer #1: Introduction:

Researchers continuously explore the utilization of artificial intelligence in

Education -> small letter in ‘education’

Jia, S. -> can you double check the citation requirement of this journal? I think most journal they only need the last name but not the initial of the first name. If initial such as , S. is not needed, please correct it throughout the manuscript.

The model was built using a variety of students’ data sets over three academic years. Through the analysis, the passing rate of the final practical examination has been improved, from about 60% to more than 80%............. I don’t understand, through what analysis and what is the important issue for this analysis? Just the observation of 60% to 80%? Or they identified the underlying reasons and therefore they have applied proper intervention to improve it? If just observation of the % change, it will become nothing special here.

Gunawardena, E. [22] used a multiple regression model to predict final exam scores for students in undergraduate business statistics courses. 366 samples, in the winter semester of 2017. Contains three predictors, namely two classroom tests and homework scores. It is proposed that the final exam as a dependent variable is affected by homework and classroom test factors. …… some incomplete sentences inside, please amend

There are currently not many articles that focus on the application of interpretable

machine learning methods in the field of educational teaching. Some research methods rely on fitting student behavior data into the model, while overlooking the

comprehensive impact of teaching methods data on the model output………..Please make some citations and give some examples to support what you claimed

In the introduction section, all the examples and citations did not provide the nationality / location of the previous studies. I think it is also important to tell the readers the country source of those subjects or studies as the cultural differences between Asia, Australia, Western countries can have a tremendous impact on why some are effective and why some are not effective in enhancing students' learning willingness. If not mentioned, it seems we have made an assumption that all these are the same in all countries in the world but it is not the case in reality.

In table 1, the left hand side is the type of predictors and outcome variable which are the most important key in this study. However, the introduction section only provided some brief and overall background information. Please write something or relevant theory behind to support why “student behavioral data indicators” or its components are potentially important to learning motivation? Similarly why teaching environment and teaching methods are also potentially important to students learning motivation. Please use some proven theories to support if possible. Otherwise, the use of these indicators or predictors will sound intuitive.

Section 2

Categorizing students' actual final exam scores into three groups (0, 1, 2) based

on their final exam scores, corresponding to failing, good and excellent grades

respectively and labeled as "low," "medium," and "high," provides a common measure of theoretical course outcomes and authentically reflects student motivation…………. Please provide some citations and explanations on why students exam performance is equivalent to learning motivation? How about some students who are slow learner that they have put tremendous efforts but only got average or below average results?

This paper uses the real data recorded by micro-assisted teaching platform in the

daily teaching process of the course "Computer Systems" for students majoring in

Software Engineering at Information Engineering School in 2021, 2022, and 2023. Thereason for choosing this course as a representative is that this course is a typical theoretical course with very little hands-on experiments, and it focuses more on the explanation and learning of theories. Therefore, the study of the relationship between

the data of this course and students' willingness to learn is representative……………If so, it also means that the subject discipline such as theory based like accounting or computer science, practical based such as sports and physical education can be very different? This should also be at least lightly mentioned in the introduction and here you may want to say to minimize unknown confounding variables or noise affecting the prediction accuracy rather than “Therefore, the study of the relationship between the data of this course and students' willingness to learn is representative”

I think there is one major limitation of this study that you can only make an assumption exam result = students’ learning motivation but it may be partially true. You should have another variable called “self-efficacy of learning motivation” so that students’ self-rate their learning motivation. If you can point out the very high correlation between the self-rate / self-efficacy one with the exam result, it will be more persuasive to use the exam result high medium and low to indicate high, middle and low motivation of students. Please highlight this in the discussion as your limitation

The section of XGBoost Model is good in explaining the model logic and formula things from a technical point of view. However, if this paper is more on the education side rather than computer science side, I suggest on top of these explanation, you can add few example from previous literature how XGBoost classifier successful do prediction in education or relevant areas so that readers can understand it may be good fit to certain kind of data structure / nature.

SHAP section – please add at least one more citation to support its usefulness

Please tell the readers if any data pre-processing, feature selection procedures (e.g. RFE or MRMR) used and any hyper-parameter tuning (e.g. gridsearch or random search?).

Please give either a very very brief explanation on Ai metrics such as F1 score, confusion matrix…etc. with at least 1 citation

Why 7:3 split but not common one 8:2 ? (and how many training and how many testing set in total?)

The section below figure 4 explaining the SHAP is very detailed and technical. It can be more precise instead of an article teaching people the interpretation of SHAP values in detail.

The whole discussion only focused on repeating the description of the results such as which one is more important which one is less. However, it lacks an in-depth discussion on WHY homework assignments, WHY discussion activities, and why game-based or storytelling things from the education point of view. As the educational-related manuscript or paper rather than a computer science or visualization paper, the discussion section should give more insight and be based on previous educational literature to make comparisons and propose explanations as well as provide future study direction. Now it sounds like a paper teaching the interpretation of data visualization especially SHAP rather than a true education related paper.

Reviewer #2: The paper is well-written in good English. The problem this paper addresses is relevant. The authors use an interesting and promising method. They present a brief literature review that is comprehensive enough to understand the main topic of the paper. The methodology appears feasible and is conducted appropriately. However, I have some comments, which I will specify alongside the relevant text when necessary.

1) It is not clear in the introduction which are the primary models for the integration of AI in teaching.

“A comprehensive analysis of educational artificial intelligence, as outlined in literature, has identified three primary models for the integration of artificial intelligence in teaching…”

2) Consider using 'opaque box' instead of 'black box' to describe the model, as it avoids potential negative connotations and supports more inclusive language

3) The use of "On the other hand" seems inappropriate as it introduces a new idea that does not contrast with the previous one.

“On the other hand, to achieve the ideal learning outcomes, teachers have tried various teaching methods to promote students' willingness to learn, such as using game-based methods…"”

4) The indicators should not be considered part of the main contributions as they were not formalized (maybe beacuse may not be necessary). They could be discussed as a marginal contribution in the discussion section. It is also important to clarify the period used to create frequency variables. Weeks? Months? Years?

5) This statement requires a reference.

“Because final exam score are the most intuitive reflection of a student's willingness to learn a course...”

6) The course chosen for analysis may not be representative due to its theoretical difficulty and low pass rate. Suggest changing the text to present it as a good option to be a starting point, with future work needed to use other courses for comparison and better external validation

7) What problems? Are they related to the limited interpretability as XGBoost feature importance only provides scores rather than detailed explanations?

“To address the problem of interpretability associated with feature importance in XGBoost,...”

8) Local explanations may be unnecessary for achieving the paper's goals as outlined in the first chapter. The section related to the local explanations could be omitted since the paper initially seeks global explanations.

9) Why did the authors arbitrarily change the model parameters that had been adjusted in advance through a cross-validation process? What is the practicality of exploring alternative parameter settings to compare model output? In other words, since the model parameters were properly adjusted earlier, exploring results with different parameter settings seems impractical and unnecessary. If the intent is to demonstrate model robustness and generalizability, simply changing parameters arbitrarily seems insufficient. A systematic approach would be more appropriate. I recommend omitting this part or explaining better its intention and need.

“The XGBoost model mentioned earlier achieved an accuracy of 85.19%. To validate the effect of adjusting the primary parameter range on accuracy, the optimal parameters were adjusted accordingly, such as the learning rate was decreased from 0.3 to 0.1...”

10) Replace Figure 7 with a table.

11) Comparing feature importance rankings from XGBoost directly with SHAP explanations is promising, as they are computed differently. Within this context, similarities could provide more evidence about the true role of variables in the data-generating process. This aspect could be more detailed and more explored in the text.

12) Use formal tables instead of figures to present model outputs such as in Fig. 7 and 9

13) In the conclusion, highlighting the six most important variables is excessive. The models seem to use only four variables for predictions, as indicated by the first Global SHAP explanations and feature importance rankings in the first model. The other variables have marginal explanation power and their ranking could be only due to chance.

14) The variables have different names throughout the text and in the figures. Please standardize them.

Reviewer #3: This manuscript contributes to education by providing a machine learning (ML) tool to predict students' willingness to learn and analyze the influence of various factors on this willingness. Please find some suggestions for the authors to consider in improving the manuscript in the reviewer attachment file.

This manuscript contributes to education by providing a machine learning (ML) tool to predict students' willingness to learn and analyze the influence of various factors on this willingness.

Below are some suggestions for the authors to consider for improving the manuscript: 1) Overall structure of the manuscript:

- Please consider restructuring the article according to the content sections like the

normal structure of an article. For example, section 3. (“Comparison and Evaluation”) and section 4. (“Explanation and Testing”) seem to belong to the Results section.

- Section 2. (“Research framework, Datasets and Methods”) should be renamed to “Methodology” to encompass all the relevant content.

- Please clearly define research questions. Subsequently, align the Methodology, Results, and Discussion sections to address these research questions.

2) Formating of manuscript:

- Please ensure consistency in the formatting of titles, images, and tables throughout

the manuscript.

- Some images such as Fig. 7 and Fig. 9 need to be considered for more appropriate

expression.

3) Research frameworks:

- Please clarify the rationale behind considering the "final exam score as the most intuitive reflection of students' willingness to learn a course."

- Please provide a clear basis for the research framework. Consider visualizing the role of each model in the procedure, specifying the input and output for each model. Fig. 1 appears confusing as it shows all factors and output labels as input sources of the models; a more detailed description of each element within the framework is needed.

4) Datasets, consider to provide details about the number of samples used for training and testing in this section (currently mentioned only in the results interpretation).

5) Methods: As stated above, please consider adjusting the structure of this section to clearly outline each method used to answer each research question.

 6) Results:

- The structure of section 3. (“Comparison and Evaluation”) might be confusing. It

would be better to first evaluate the performance of the XGBoost model and then compare it with other models (SVM, DT, RF, and ANN). Also, detail the validation method used to evaluate the models' performance.

- For Tabs. 3, please provide some experimental values to ensure the values are optimal, potentially through ablation studies.

- For Fig. 2, based on the matrix, please explain why the error rate is higher for high true labels compared to lower labels.

- Please provide a more detailed analysis of the results shown in Fig. 7 to 9 to support your claims.

6. PLOS authors have the option to publish the peer review history of their article (what does this mean? ). If published, this will include your full peer review and any attached files.

**Do you want your identity to be public for this peer review?** For information about this choice, including consent withdrawal, please see our Privacy Policy .

Reviewer #1: **Yes: ** Indy Man Kit Ho

Reviewer #2: **Yes: ** Rogério Luiz Cardoso Silva Filho

Reviewer #3: No

---

## [Author Response · Author response to Decision Letter 1]

18 Aug 2024

Dear Reviewers,

Thank you for your valuable comments. Based on your feedback, we have made the necessary modifications to our manuscript. We hope these changes align with the publication standards of this journal. A point-by-point response has been uploaded in the Word file named response_to_reviewers.docx.

Best regards,

Authors

---

## [Decision Letter · Decision Letter 1]

22 Oct 2024

PONE-D-24-23837R1What Factors Enhance Student engagement/achievement? A Machine Learning and Interpretable Methods ApproachPLOS ONE

Dear Dr. Mao,

Thank you for submitting your manuscript to PLOS ONE. After careful consideration, we feel that it has merit but does not fully meet PLOS ONE’s publication criteria as it currently stands. Therefore, we invite you to submit a revised version of the manuscript that addresses the points raised during the review process.

While the revisions made have addressed some of the initial concerns, two of the reviewers pointed out that further refinements are necessary. Specifically, the link between student engagement and achievement needs further clarification, and improvements to the manuscript’s structure and writing style are required to enhance its overall clarity and accessibility.

We look forward to receiving your revised manuscript.

Kind regards,

Leonard Moulin

Academic Editor

PLOS ONE

Journal Requirements:

Reviewers' comments:

Reviewer's Responses to Questions

**Comments to the Author**

1. If the authors have adequately addressed your comments raised in a previous round of review and you feel that this manuscript is now acceptable for publication, you may indicate that here to bypass the “Comments to the Author” section, enter your conflict of interest statement in the “Confidential to Editor” section, and submit your "Accept" recommendation.

Reviewer #1: (No Response)

Reviewer #2: All comments have been addressed

Reviewer #3: All comments have been addressed

Reviewer #4: (No Response)

2. Is the manuscript technically sound, and do the data support the conclusions?

Reviewer #1: Partly

Reviewer #2: Yes

Reviewer #3: (No Response)

Reviewer #4: Partly

3. Has the statistical analysis been performed appropriately and rigorously? 

Reviewer #1: Yes

Reviewer #2: Yes

Reviewer #3: Yes

Reviewer #4: No

4. Have the authors made all data underlying the findings in their manuscript fully available?

Reviewer #1: Yes

Reviewer #2: Yes

Reviewer #3: Yes

Reviewer #4: Yes

5. Is the manuscript presented in an intelligible fashion and written in standard English?

Reviewer #1: No

Reviewer #2: Yes

Reviewer #3: Yes

Reviewer #4: No

6. Review Comments to the Author

Reviewer #1: Thanks for the great effort in making the revision.

Some issues want a bit further fine tuned and revision:

Line 26 to 27 - the teaching process of pure theoretical course is rather boring. Therefore this paper takes the teaching data of the most boring purely theoretical course...............To say pure theory course as most boring can be dogmatic and it does not sound nice. I recommend "Due to the lack of practical components in the teaching and learning processes, the learning interaction and student engagement activities of these theory-based courses such as "Computer System" can be greatly limited. Our results showed that: actively taking.......................

Similarly revise line 194 to 199 and line 237 to 242 so that we won't be too subjective to say boring

Line 88 - may delete "2017", sounds a bit weird

Line 68 to 102 - You have used many citations to support the strong association between students' engagement and academic performance/achievement, can you highlight a bit the findings (e.g. correlation ?) from those studies? You may use for example (r=0.90) or R2 values if it was regression, to supplement the findings from previous studies when you highlighted those variables. I think this is quite important as you have made an assumption exam results ~ engagement. You need to show the very high validity between these or otherwise the topic and study findings can no longer be valid as you can't use exam result to reflect students engagement anymore.

Line 184 - means  reflects

Line 222 to 229 - i think you should use past tense throughout?

Line 249 - 314 (70%)..........135 (30%)

Line 472 - reprsent

Overall speaking, the modeling, data processing and technical things are clear and good.

I think the study itself also provides good insights to machine learning practitioners and also educators especially the SHAP interpretation (although quite clumsy and not smartly written). However, the writing of this manuscript can be further enhanced (e.g. by a native English professional writer). Although most or nearly all sentences are readable, the paragraphs arrangement, sentence structure and writing style are somewhat clumsy, with sentences redundant or duplicated sometimes. I did never see a conclusion with more than 1 page in other manuscripts (if so, it is either not precise and well written enough or some contents should be in discussion rather than conclusion). Meanwhile some contents or similar sentences existed in many places repeatedly in introduction, method/results and discussion/conclusion which are not necessary....as it finally made your manuscript necessarily long (732 lines now). Therefore it is recommended to further enhance the writing style before finally getting accepted.

Reviewer #2: (No Response)

Reviewer #3: (No Response)

Reviewer #4: The paper presents several challenges in its current form, which detract from its clarity. There are instances where the argumentation appears confused, making it difficult for readers to follow the line of reasoning.

The link between achievement and engagement is insufficiently developed and explained, weakening the argument's depth and making it hard to understand how these two variables relate in the context of the study.

Furthermore, some machine learning concepts are not explained in enough detail for readers who may be unfamiliar with the field, limiting the paper’s accessibility to a broader audience.

Additionally, an ordinal response variable is treated as a categorical one, which could lead to misinterpretation of the data and undermines the validity of the results.

7. PLOS authors have the option to publish the peer review history of their article (what does this mean? ). If published, this will include your full peer review and any attached files.

**Do you want your identity to be public for this peer review?** For information about this choice, including consent withdrawal, please see our Privacy Policy .

Reviewer #1: **Yes: ** Indy Man Kit HO

Reviewer #2: **Yes: ** Rogerio Luiz Cardoso Silva Filho

Reviewer #3: No

Reviewer #4: No

---

## [Author Response · Author response to Decision Letter 2]

10 Dec 2024

Thank you for your valuable comments and suggestions. We have carefully addressed all the feedback provided. The response to the reviewer’s comments is attached in a Word file, along with the revised manuscript for your review.

---

## [Decision Letter · Decision Letter 2]

2 Jan 2025

PONE-D-24-23837R2What Factors Enhance Student achievement? A Machine Learning and Interpretable Methods ApproachPLOS ONE

Dear Dr. Mao,

Thank you for submitting your manuscript to PLOS ONE. After careful consideration, we feel that it has merit but does not fully meet PLOS ONE’s publication criteria as it currently stands. Therefore, we invite you to submit a revised version of the manuscript that addresses the points raised during the review process.

The reviewers have provided detailed and constructive feedback on the manuscript, identifying several areas that require significant revision before the manuscript can be considered for publication, particularly methodological concerns regarding the treatment of the ordinal response variable. Thank you.  Musa Adekunle Ayanwale

Academic Editor

PLOS ONE Be sure to:

Indicate which changes you require for acceptance versus which changes you recommendAddress any conflicts between the reviews so that it's clear which advice the authors should followProvide specific feedback from your evaluation of the manuscript

We look forward to receiving your revised manuscript.

Kind regards,

Musa Adekunle Ayanwale

Academic Editor

PLOS ONE

Reviewers' comments:

Reviewer's Responses to Questions

**Comments to the Author**

1. If the authors have adequately addressed your comments raised in a previous round of review and you feel that this manuscript is now acceptable for publication, you may indicate that here to bypass the “Comments to the Author” section, enter your conflict of interest statement in the “Confidential to Editor” section, and submit your "Accept" recommendation.

Reviewer #1: All comments have been addressed

Reviewer #2: All comments have been addressed

Reviewer #3: All comments have been addressed

Reviewer #4: (No Response)

2. Is the manuscript technically sound, and do the data support the conclusions?

Reviewer #1: Yes

Reviewer #2: Yes

Reviewer #3: Yes

Reviewer #4: Partly

3. Has the statistical analysis been performed appropriately and rigorously? 

Reviewer #1: Yes

Reviewer #2: Yes

Reviewer #3: Yes

Reviewer #4: No

4. Have the authors made all data underlying the findings in their manuscript fully available?

Reviewer #1: Yes

Reviewer #2: Yes

Reviewer #3: Yes

Reviewer #4: Yes

5. Is the manuscript presented in an intelligible fashion and written in standard English?

Reviewer #1: Yes

Reviewer #2: Yes

Reviewer #3: Yes

Reviewer #4: No

6. Review Comments to the Author

Reviewer #1: Thanks for the great efforts in revising the manuscript. The readability and language have been greatly enhanced. Very few amendments in the introduction are recommended.

Line 76 - 107: When using r value the correlation coefficient to support e.g. the academic achievement is associated with the engagement, we need to pay attention to the actual r value. Even though it is statistical significant such as p < 0.05, but if the r value is just 0.1-0.4, we should not say it is strong association, correlation or connection. For those value below 0.3, I may not present them in the introduction. Even 0.3-0.5 only indicates moderate correlation/association but still better than 0.1-0.29. As you want to tell the readers that they have some relationship or good association but the findings from previous studies showing r = 0.17, 0.28, 0.14, 0.24....etc. These on the other hand are somewhat contradicting to what you said. We may need to tell the truth in a strategic and smart way. E.g. Previous studies showed the association between the academic performance and student engagement through the student behavior (i.e. "Distinct days active" and "Total number of posts made in discussion forums") of MOOCs platform (r=0.32 to 0.38) [21].

Similarly, please review line 102-106, I will only choose the amount of work done r = 0.30 (2 significant difference) but not to show others with r value <0.3 as the correlations were so weak to support your statement/arguments.

Line 85 - 90: The effect size using Cohen's d is to assess the magnitude of difference between two groups. It is NOT an assessment of the correlation. Therefore, it is a flaw or misunderstanding to say using Cohen's d to show a positive linear relationship. Moreover, I have checked the paper by Phebe [23] that it does not have the d value 2.05 very large effect. Please check carefully and also clarify this part.

Reviewer #2: The authors have made substantial improvements to transform the original manuscript into its current version. The two main weaknesses of the original paper - the relationship between engagement and achievement, and the writing style - have been effectively addressed. While the engagement-achievement relationship may remain a point of discussion in the field, the authors have now provided enough support for this connection throughout the text. The minor issues have also been resolved, and the paper is now ready for submission.

Reviewer #3: The author has provided detailed feedback and tried to fully meet each reviewer's request. They provided reasonable arguments to explain the adjustments and additions, helping to clarify the points that the previous manuscript was not complete.

The integration of the XGBoost model and the SHAP method in the study is also emphasized and demonstrated through specific charts and figures, helping to increase persuasiveness.

Reviewer #4: Thank you for the opportunity to review this manuscript. Below, I provide my comments, with particular emphasis on a major issue regarding the treatment of the model output, which I believe needs to be addressed before proceeding further with this work.

The primary issue with this manuscript lies in the treatment of the ordinal response variable as categorical. This is a critical methodological flaw that undermines the reliability and interpretability of the results. To address this:

1. I strongly recommend exploring models specifically designed for ordinal data, such as ordinalForest (refer to: https://link.springer.com/article/10.1007/s00357-018-9302-x)

2. Include comparison indices suitable for ordinal responses to ensure the analysis is rigorous and appropriate for the data type.

examples:

- Ballante, E., Figini, S., and Uberti, P. (2022). A new approach in model selection for ordinal target variables. Computational Statistics, 37(1):43–56.

- Cardoso, J. S. and Sousa, R. (2011). Measuring the performance of ordinal classification. International Journal of Pattern Recognition and Artificial Intelligence,25(08):1173–1195.

In my opinion, until this fundamental concern is addressed, the validity of the findings remains questionable.

General Comments

1. The manuscript would benefit significantly from a professional proofreading service to improve the English language and ensure better fluidity.

2. Several sections require greater clarity and coherence to enhance readability and logical flow.

Additional Comments

Abstract

• The opening sentence is unclear and needs to be rewritten for better readability and impact.

• From line 22, starting with “The results show…,” restructure this section to present the methodology first, followed by the results. The current arrangement is confusing, as it alternates between these elements.

Literature Review

• Specify the type of courses being analyzed (e.g., in-person, online, blended) and the educational level (e.g., K-12, university).

Chapter 1.2.1

• Following revisions, the focus of the study now appears to be on "student achievements." Revise this section to better align with the updated focus. Is it necessary all the discussion?

Chapter 1.4

• Expand the explanation of the “opaque box” and provide supporting citations.

Chapter 1.5

• Clearly specify the characteristics of the course: Is it in-person or online? What is the educational level? How are examinations conducted?

Methodology

1. Add citations when introducing machine learning methods.

2. Justify the division of the dataset into 7:3 or 8:2 splits with appropriate references.

3. Specify the grading scale of the test (e.g., 0 to 100).

4. Provide detailed explanations of the indices used for model comparison (e.g., accuracy, F1 score).

5. On line 402, when discussing “Traditional feature importance…,” include citations to strengthen this point.

Discussion

• Relate the results more explicitly to existing literature to better contextualize the findings and strengthen the paper’s contribution.

I hope these comments provide clear guidance for improving the manuscript. Please feel free to contact me for any further clarification.

7. PLOS authors have the option to publish the peer review history of their article (what does this mean? ). If published, this will include your full peer review and any attached files.

**Do you want your identity to be public for this peer review?** For information about this choice, including consent withdrawal, please see our Privacy Policy .

Reviewer #1: **Yes: ** Indy Man Kit HO

Reviewer #2: **Yes: ** Rogerio Luiz Cardoso Silva Filho

Reviewer #3: No

Reviewer #4: No

---

## [Author Response · Author response to Decision Letter 3]

6 Feb 2025

Please find the attached word document for the detailed reviewer's response statements from the authors.

Thank You

---

## [Decision Letter · Decision Letter 3]

19 Mar 2025

PONE-D-24-23837R3What Factors Enhance Student achievement? A Machine Learning and Interpretable Methods ApproachPLOS ONE

Dear Dr. Mao,

Thank you for submitting your manuscript to PLOS ONE. After careful consideration, we feel that it has merit but does not fully meet PLOS ONE’s publication criteria as it currently stands. Therefore, we invite you to submit a revised version of the manuscript that addresses the points raised during the review process. In the abstract, clearly state the research gap in a sentence, include key results, and refine the conclusion to highlight major findings and implications. The introduction should better define the study focus, objectives, and methodology for improved reader engagement. Citation inconsistencies should be corrected by adhering to a single referencing style. Additionally, minor grammatical errors and better linkage in the literature review will improve coherence. Musa Adekunle Ayanwale

Academic Editor

PLOS ONE

We look forward to receiving your revised manuscript.

Kind regards,

Musa Adekunle Ayanwale

Academic Editor

PLOS ONE

Journal Requirements:

Reviewers' comments:

Reviewer's Responses to Questions

**Comments to the Author**

1. If the authors have adequately addressed your comments raised in a previous round of review and you feel that this manuscript is now acceptable for publication, you may indicate that here to bypass the “Comments to the Author” section, enter your conflict of interest statement in the “Confidential to Editor” section, and submit your "Accept" recommendation.

Reviewer #5: All comments have been addressed

Reviewer #6: All comments have been addressed

2. Is the manuscript technically sound, and do the data support the conclusions?

Reviewer #5: Yes

Reviewer #6: Partly

3. Has the statistical analysis been performed appropriately and rigorously? 

Reviewer #5: Yes

Reviewer #6: Yes

4. Have the authors made all data underlying the findings in their manuscript fully available?

Reviewer #5: Yes

Reviewer #6: Yes

5. Is the manuscript presented in an intelligible fashion and written in standard English?

Reviewer #5: Yes

Reviewer #6: Yes

6. Review Comments to the Author

Reviewer #5: (No Response)

Reviewer #6: The manuscript contains some grammatical errors, citation inconsistencies, and insufficient linkage in the literature review. Please implement the corrections provided in the reviewer upload.

Best of luck, and great work!

7. PLOS authors have the option to publish the peer review history of their article (what does this mean? ). If published, this will include your full peer review and any attached files.

**Do you want your identity to be public for this peer review?** For information about this choice, including consent withdrawal, please see our Privacy Policy .

Reviewer #5: **Yes: ** Oluwaseyi Aina Gbolade Opesemowo

Reviewer #6: **Yes: ** Victor Eyo Essien

---

## [Author Response · Author response to Decision Letter 4]

29 Mar 2025

Dear Editor and Reviewers,

Thank you for your valuable comments. Based on your feedback, we have made the necessary modifications to our manuscript. We hope these changes meet the publication standards of this journal.

Best regards,

Authors

Editor Comments:

1.In the abstract, clearly state the research gap in a sentence, include key results, and refine the conclusion to highlight major findings and implications.

AU comments: Thank you for your valuable suggestions! We have revised the abstract to prominently address four key aspects: the research gap, main methods, major findings, and implications (see lines 16–30).

2.The introduction should better define the study focus, objectives, and methodology for improved reader engagement.

AU comments: We appreciate this feedback. The introduction has been restructured to clearly outline the study’s focus, objectives, and methodology (lines 35–71).

3.Citation inconsistencies should be corrected by adhering to a single referencing style.

AU comments: hank you for noting this. We have standardized all citations to a single referencing style, with modifications highlighted in yellow in the Reference section.

4.better linkage in the literature review will improve coherence

AU comments: We greatly appreciate this suggestion. The literature review has been revised to strengthen logical connections between sections (see lines 134–135, 150–153, and 155–157).

5.minor grammatical errors

AU comments: Thank you for highlighting this. We have thoroughly proofread the manuscript to correct grammatical errors, with revisions marked in yellow.

Reviewer #6:

1. The manuscript contains some grammatical errors.

AU comments: Thank you for your observation. We have carefully reviewed the entire text and corrected grammatical issues, with changes highlighted in yellow.

2.Citation inconsistencies.

AU comments: We appreciate your attention to this detail. All references now adhere to a consistent style, as indicated in the yellow-highlighted modifications.

Insufficient linkage in the literature review.

AU comments: Thank you for this insightful suggestion. We have enhanced the flow and connectivity of the literature review (see lines 134–135, 150–153, and 155–157).

---

## [Editor Report · Decision Letter 4]

8 Apr 2025

What Factors Enhance Students’ achievement? A Machine Learning and Interpretable Methods Approach

PONE-D-24-23837R4

Dear Dr. Hui Mao,

We’re pleased to inform you that your manuscript has been judged scientifically suitable for publication and will be formally accepted for publication once it meets all outstanding technical requirements.

Kind regards,

Musa Adekunle Ayanwale

Academic Editor

PLOS ONE

Additional Editor Comments (optional):

I have carefully reviewed the updated manuscript and am pleased to note that all reviewers’ comments have been thoroughly and thoughtfully addressed. Given these improvements, I am satisfied that the manuscript meets PLOS ONE’s publication standards. Thank you.
---

## [Editor Report · Acceptance letter]

PONE-D-24-23837R4

PLOS ONE

Dear Dr. Mao,

I'm pleased to inform you that your manuscript has been deemed suitable for publication in PLOS ONE. Congratulations! Your manuscript is now being handed over to our production team.

Kind regards,

on behalf of

Dr. Musa Adekunle Ayanwale

Academic Editor

PLOS ONE